# Deimplementation strategy to reduce overtreatment of asymptomatic bacteriuria: a study protocol for a stepped-wedge cluster randomised trial

Tessa MZXK van Horrik [1], Suzanne E Geerlings,[1] Janneke E Stalenhoef,[2] Cees van Nieuwkoop,[3] Joppe B Saanen,[4] Caroline Schneeberger,[5] Bart J Laan [1]

For numbered affiliations see end of article.

**Correspondence to**
Tessa MZXK van Horrik;
t.m.vanhorrik@amsterdamumc.nl

## ABSTRACT

**Introduction** Antimicrobial treatment of asymptomatic bacteriuria (ASB) is one of the most common unnecessary uses of antimicrobials. Earlier studies have shown that the prevalence of this inappropriate treatment ranges from 45% to 83%. Multifaceted interventions based on international guidelines and antimicrobial stewardship can decrease overtreatment of ASB. We have designed a study protocol with the main objective of reducing overtreatment of ASB by 50% through use of a deimplementation strategy.

**Methods and analysis** We will use a stepped-wedge cluster randomised design, comparing outcomes before and after introduction of our intervention in the emergency department (ED) of five hospitals (clusters) in the Netherlands. All patients (≥18 years old) who have a urine test performed in the ED will be screened for eligibility. The deimplementation strategy consists of a combination of interventions, including education, audit and feedback. The primary endpoint is overtreatment of ASB in patients without risk factors (eg, pregnancy, planned invasive urological procedures and neutropenia). Secondary endpoints are the duration of antimicrobial treatment for ASB, the number of urine cultures and urinalysis per 1000 patients, and overtreatment of positive urinalysis in asymptomatic patients.

**Ethics and dissemination** Ethical approval was obtained from the medical ethics research committee of the Academic Medical Centre (Amsterdam, the Netherlands) with a waiver for informed consent. Local feasibility was obtained by the local institutional review boards of all participating hospitals. Our study aims to reduce inappropriate screening and treatment of ASB in EDs, improve healthcare quality, lower the increase in antimicrobial resistance and save costs. If proven (cost)-effective, this study provides a well-suited strategy for a nationwide approach to reduce overtreatment of ASB. Relevant results of our study will be disseminated through publications in peer-reviewed journals and presentations at relevant (scientific) conferences.

**Trial registration number** NL8242; Pre-results.

## Strengths and limitations of this study

► We will use a stepped-wedge cluster randomised design, ultimately resulting in implementation of our intervention in all participating hospitals.
► Our strategy is well suited for broad-scale deimplementation in emergency departments.
► A possible limitation of this study is that we are not able to evaluate the impact of an individual intervention since we are using a multifaceted deimplementation strategy.
► Another possible limitation is that we are limited by documentation in the patient files, and therefore, we might overestimate the number of patients with asymptomatic bacteriuria.

## INTRODUCTION

There is a growing urgency to improve healthcare quality and save costs by implementing practices to reduce unnecessary diagnostics, diagnostic error and related overtreatment, so-called low-value care practices. This is exemplified by international campaigns, such as Choosing Wisely and Do not Do prompts, which were launched to reduce unnecessary care in several medical services, resulting in less risks for the patients, saving costs and increasing value for care.[1 2] A parallel campaign initiative titled 'To Do or Not to Do?' was introduced to deimplement unnecessary care in the Netherlands.[3]

Antimicrobial resistance, as a result of antibiotic overconsumption, is a global threat to public health, and there is global consensus about reducing the use of antibiotics.[4] Urinary tract infections (UTIs) are one of the most common categories of infections for which antimicrobials are prescribed.[5] However, not all antimicrobial prescriptions are necessary. This is especially true for asymptomatic bacteriuria (ASB), which is the presence of bacteria in the urine of a patient who does not have symptoms of a UTI. ASB is a common finding, especially among women, elderly and patients with urinary catheters. The prevalence of ASB

in adults over 65 years is high since the prevalence of ASB increases with age.[6] Guidelines strongly recommend not to screen for or treat ASB with antimicrobials, except for specific patients at risk of developing a complicated UTI.[5] This is also outlined in the American Choosing Wisely campaign, which judges treatment of ASB as one of five low-value care services that should be avoided.[7] In 2015, this resulted in the recommendation 'Do not use antimicrobials to treat bacteriuria in older adults unless specific urinary tract symptoms are present'.[7]

Regardless, antimicrobials are still frequently used for patients with ASB. A systematic review and meta-analysis from 2017 showed a pooled prevalence of 45% among 4129 ASB cases (95% CI 39 to 50) who received inappropriate antimicrobial treatment.[8] The prevalence was even higher in a recent retrospective evaluation of 25 hospitals in the USA, in which 64% of the 2225 positive urine cultures were classified as ASB, and 72% of 961 patients with ASB were treated with antimicrobials.[9] The exact use of antimicrobials for ASB in hospitals in the Netherlands is unknown. However, in 2015, a prospective study performed in 10 nursing homes in the Netherlands showed that 115 (32%) of 356 residents with a possible UTI were treated inappropriately, and treatment of ASB was the most common reason.[10] These results indicate that reducing the treatment of ASB is an appropriate target of antimicrobial stewardship strategies.

Earlier studies demonstrated that multifaceted interventions can effectively reduce overtreatment of ASB in nursing home residents and patients with urinary catheters. In 2005, a cluster randomised trial in 12 nursing homes in Canada and the USA showed a reduction in antimicrobial use after the multifaceted implementation of a diagnostic and treatment algorithm (1.17 vs 1.59 courses of antimicrobials/1000 resident days).[11] For catheterised patients in the USA, a controlled before–after study in 2015 showed a decrease of urine culture ordering from 41.2 to 23.3 per 1000 bed-days.[12] Furthermore, short educational sessions combined with feedback in two academic tertiary acute care hospitals in Canada, performed in 2015, resulted in a reduction of overtreatment for ASB of 8% (2 of 24 patients) in the intervention group compared with 52% (14 of 29 patients) in the control group (OR 0.1, 95% CI 0.02 to 0.49).[13] In 2018, these findings resulted in the development of an implementation guide to reduce overtreatment of ASB.[14]

However, effectiveness studies after the launch of this implementation guide have not yet been performed in emergency departments (EDs). The results of earlier studies suggest that improving ASB management in the ED is necessary. In 2017, a prospective observational study performed in the ED of a tertiary care centre in the USA showed that 27 of 71 (38%) older adults were not correctly diagnosed with a UTI.[15] Furthermore, overtreatment of a suspected (but unconfirmed) UTI was very common in a retrospective study performed in the ED of another tertiary care centre in the USA in 2013, where 63 of 66 (95%) patients were treated while they had a negative urine culture.[16]

In this study, our aim was to reduce overtreatment of ASB by 50% by using a multifaceted deimplementation strategy based on the implementation guide described previously.

## METHODS AND ANALYSIS

### Study design and setting

For the development of our study protocol, we used the reporting guidelines for stepped-wedge cluster randomised trials.[17] We will use a repeated measurement stepped-wedge cluster randomised design. The deimplementation strategy will be introduced in the ED of five hospitals (clusters) in the Netherlands (one university and four general teaching hospitals) in six time periods consisting of 1 or 2 months, depending on how many patients will be included in the first month. Therefore, the total study duration is 6–12 months (figure 1). The order in which the participating hospitals will receive the intervention will be randomly determined by an independent data manager. Every cluster will start in control condition, and by the end of the trial, every cluster will have received the intervention. We planned to start our study on 1 May 2020. However, due to the COVID-19 pandemic, the start of our study will be postponed until 1 October 2020.

### Patient selection

We will screen all patients (≥18 years old) who have urine tests (culture and/or urinalysis) that were obtained during an ED presentation (figure 2). First, we will exclude all patients who have no positive urinalysis or urine culture, that is, if both the urinalysis and culture are negative (table 1). Further exclusion criteria are patients with a symptomatic UTI (systemic or local symptoms, according to the Center for Disease Control's National Healthcare Safety Network for Urinary Tract Infection) (table 1),[18] active treatment for a UTI on presentation, patients with an alternate site of infection for which they receive antimicrobial treatment, and patients with ASB and risk factors, defined as pregnant women, patients prior to planned invasive urological procedures associated with mucosal trauma (including transurethral surgery of the prostate or bladder, ureteroscopy, including lithotripsy and percutaneous stone surgery) and high-risk neutropenia (defined as absolute neutrophil count<500 cells/µL). Patients with altered mental status (AMS) and with possible systemic symptoms of an infection and bacteriuria, since treatment with antimicrobials is then considered appropriate.

### Primary and secondary endpoints

The primary endpoint is the percentage of patients with ASB without risk factors or another alternative site of infections, who are treated with antimicrobials. We define ASB according to the international guideline provided

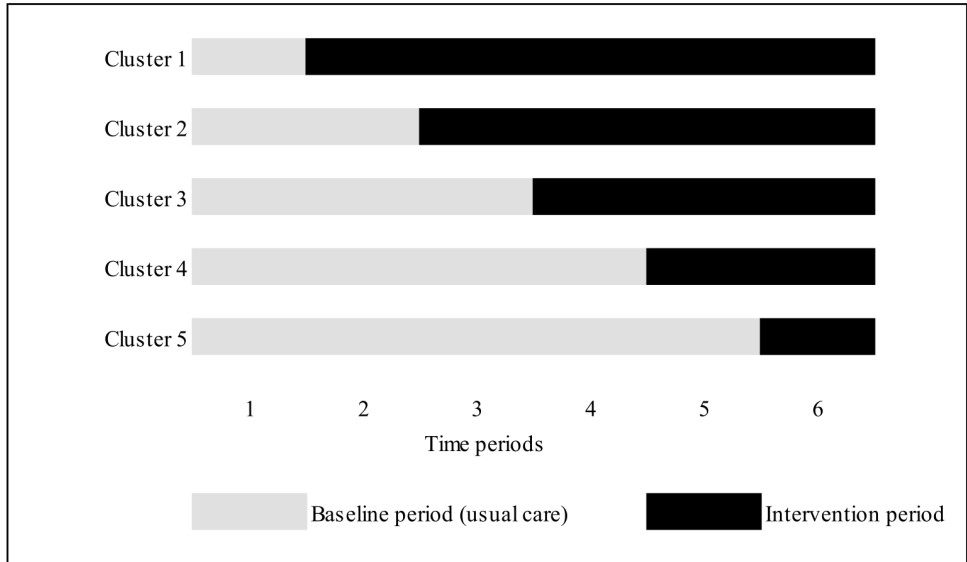

**Figure 1** Stepped wedge design for deimplementation strategy.

by the Infectious Diseases Society of America (table 1).[5] Secondary endpoints are the duration of antimicrobial treatment for ASB, the number of urine cultures and urinalysis (dipsticks and microscopic analysis) per 1000 patients at the ED, the percentage of asymptomatic patients treated for urine cultures that are considered positive in daily practice, a quantitative count of bacteria of ≥$10^3$ colony-forming units (CFU)/mL and the percentage of patients without symptoms or risk factors treated with antimicrobials for a positive urinalysis without an obtained culture. Explorative analyses will evaluate the total number of urine cultures ordered in

the hospital and indications for negative urine cultures during the baseline period.

### Data collection

The study will collate a list of all urine cultures and urinalyses performed in the ED, including corresponding microbiology laboratory and clinical chemistry data. All relevant patient information, including the presence of other possible infections, of those with positive cultures will be reviewed from medical and nursing records by a research physician to determine if patients have ASB. For patients with ASB, we will collect data on antimicrobial

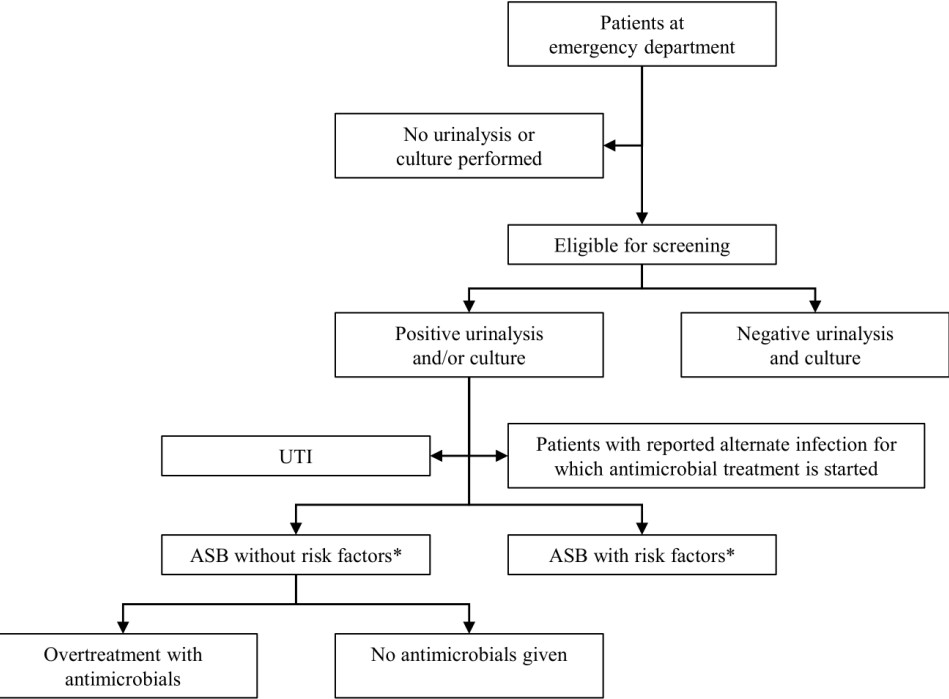

**Figure 2** Study profile. *Risk factors are pregnancy, patients prior to planned urological procedures associated with mucosal trauma and high-risk neutropenia. ASB, asymptomatic bacteriuria; UTI, urinary tract infection.

**Table 1** Definitions of positive urinalysis, UTI and ASB

| Item | Definition |
|------|------------|
| Positive urinalysis[32] | Urinalysis must have one or more of the following results:<br>Dipstick<br>► Abnormal nitrites.<br>► Abnormal leucocyte esterase.<br>Microscopic analysis<br>► >5 leucocytes/high-powered field (or reference ranges according to local standard operating procedures).<br>► Abnormal amount of bacteria according local operating procedures. |
| ASB[5] | The presence of one or more species of bacteria growing in the urine at specified quantitative counts (≥$10^5$ CFU/mL) regardless of the presence of pyuria, without UTI symptoms or signs. |
| UTI[18] | Patient has at least one of the following signs or symptoms: fever (>38°C),* suprapubic tenderness,* costovertebral angle pain or tenderness,* urinary urgency, urinary frequency or dysuria,† and patient has a urine culture with no more than two species of organisms identified, at least one of which is a bacterium of ≥$10^5$ CFU/mL. |
| Catheter-associated UTI† [18] | A UTI where a urinary catheter was in place for >2 days on the date of event, with day of placement being day 1, and was in place on the date of the event or the day before. |

*With no other recognised cause.
†A urinary catheter in situ could cause patient complaints of frequency, urgency or dysuria. Therefore, these symptoms cannot be used in patients with catheters.
ASB, asymptomatic bacteriuria; CFU, colony-forming unit; UTI, urinary tract infection.

treatment, including antimicrobial prescription and duration of antimicrobial therapy, and the specified bacterial count and pathogen according to the standard operating procedure of the microbiology laboratory. For feasibility reasons, a random sample of a total number of 220 negative urine cultures will be screened by medical record review to evaluate the indication for testing during the baseline period. We will use these data in the further development of an additional local deimplementation strategy based on the assessment of determinants of current practice. Furthermore, we will collect the patient-level variables, including patient characteristics (age, gender and residency in nursing home), Charlson Comorbidity Index,[19] Modified Early Warning Score,[20] chronic urinary catheter use, reason for presentation at the ED and use of antimicrobials for uropathogens in the last 7 days before presentation at the ED. We will also collect the reasons for urine testing documented in the medical record. Data from medical records will be collected in an electronic GCP-compliant database, and we will audit a random sample of 10% to assure the validity of data collection.

### Deimplementation strategy

Our deimplementation strategy is a tailored multifaceted intervention based on a Dutch deimplementation guide and an international implementation guide to reduce overtreatment of ASB in low-risk patients.[14 21] We will use a combination of interventions, such as education, audit and feedback and organisational adjustments, because these strategies have shown to be successful for reducing treatment of ASB in prior studies.[11 22] The deimplementation strategy can be divided into a general part that will be introduced in every hospital by our project group, and additional local strategies introduced by local healthcare workers, focused on the local setting (table 2).

For the general strategy, we will start the intervention period by holding one or more educational meetings about overtreatment of ASB for all healthcare workers in the ED. During these educational meetings, we will inform all healthcare workers about the current ASB guidelines and discuss the local barriers and facilitators to reduce its overtreatment. Next to this, the results of the baseline period will be used as competitive feedback report between the participating hospitals. This competitive feedback report will be used to raise awareness among healthcare workers.[23] During this meeting,

**Table 2** Summary of the deimplementation strategy

| Item | Description |
|------|-------------|
| **General part** | |
| Educational meeting with competitive feedback | Combinations of strategies to reduce treatment of ASB |
| Clinical decision tools | Pocket cards and posters with algorithms as an aid and reminder for appropriate and inappropriate urine testing and treatment for ASB |
| **Local part** | |
| Local champion | One of the leading physicians who is responsible for the deimplementation strategy |
| ASB team | Consists of a local champion, emergency doctor, infectious disease physician, resident internal medicine, and nurse for the emergency department. |
| Additional interventions | Based on local setting. for example, educating and/or reminding healthcare workers about appropriate urine testing and treatment of ASB. |

ASB, asymptomatic bacteriuria.

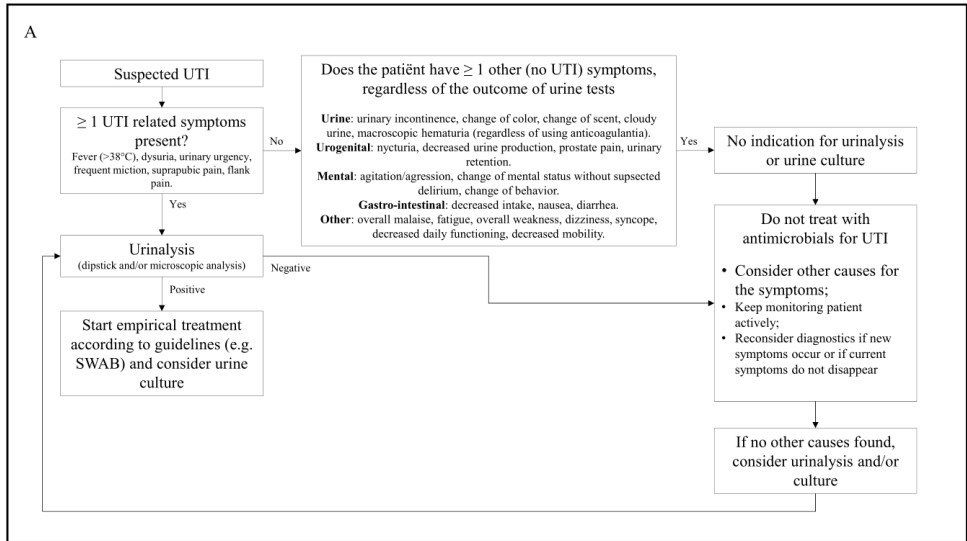

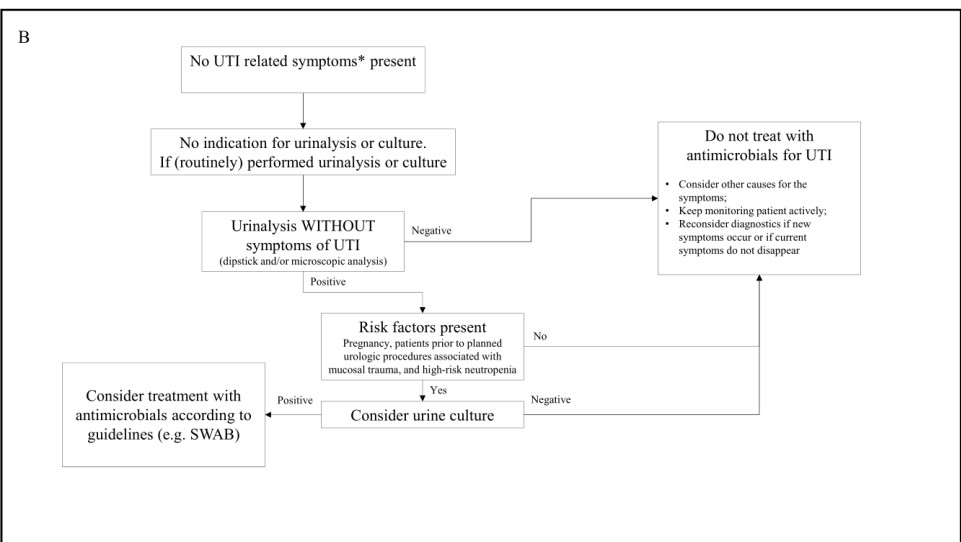

**Figure 3** (A) Algorithm of indications for urinalysis and cultures in symptomatic patients. (B) Algorithm of indications for urinalysis and cultures in asymptomatic patients. *Fever (>38°C), painful urination, urinary urgency, frequent miction, suprapubic pain, flank pain. SWAB, Stichting Werkgroep Antibiotica Beleid (the Dutch Working Party on antibiotic policy); UTI, urinary tract infection.

we will distribute posters and pocket cards with an algorithm (figure 3A,B) for appropriate indications for urine cultures and urinalysis, and overtreatment of ASB, based on current guidelines.[5 24] The kick-off meeting will be led by the project coordinator and the local champion. These tools will be available throughout the intervention period and after the intervention period, if necessary. We will send emails with the anonymous feedback report and the algorithm to healthcare workers who were not able to join the meeting.

For the additional local part of the strategy, a local champion, one of the leading physicians, will be appointed in each participating hospital and will be responsible for the deimplementation strategy. The local champion will start an asymptomatic bacteriuria quality improvement team (ASB team), which will consist of at least an emergency doctor, infectious disease physician, a resident internal medicine and a nurse from the ED. The ASB team will be motivated to assess barriers and facilitators within their hospital and to start additional interventions based on the local setting. For example, by educating and/or reminding healthcare workers about appropriate urine testing and treatment for ASB, and evaluating and/or adjusting local protocols to reduce inappropriate urine testing and antimicrobial use. Further, the ASB team will collaborate with the antibiotic stewardship teams. During

the intervention period, the project coordinator will be in contact with the local champion and/or the ASB-team once every 2 weeks to exchange ideas and experiences based on interim results (ordering patterns of urine culture) and assessment of local facilitators and barriers.

## Sample size

We will use a stepped-wedge hospital-randomised design with six time periods and five steps, in which one hospital will switch from control to treatment at each step. Previous studies have shown reduction of inappropriate treatment for ASB of approximately 50% following deimplementation strategies.[8] Therefore, we anticipate that the proportion of patients who received inappropriate antimicrobial treatment for ASB will be reduced by 50% from 0.450 to 0.225 after the deimplementation strategy. Assuming an intrahospital correlation coefficient of 0.10, a total of at least 420 patients (=14 patients×six time periods×five hospitals) will achieve 80% power to detect this reduction using the two-sided Wald Z-test with a significance level of 0.05. We used PASS V.15.0.10 to calculate this sample size. During the same time period, we will include additional patients with positive urinalyses.

## Statistical analysis

### Effect evaluation

Categorical data were summarised as frequency and percentage, and continuous data as mean (SD) or median (range). We will use fixed effects models to analyse the overall effect of our deimplementation strategy on overtreatment of ASB.[25] To adjust for clustering and temporal trends, we will include the clusters (hospitals) as a categorical predictor and the time in months as a fixed effect in the model. To correct for the level 2 variance, we will multiply the SE estimates by the square root of the design effect (DEFT). We will evaluate the differences between baseline period and intervention period according an intention to treat principle. To adjust for confounders, we will include the main risk factors for overtreatment of ASB (namely, age, acutely AMS and positive urinalysis)[26] in the model. We will use the Akaike's Information Criterion and analysis of variance tables to compare the fitting of the models. We will report the results of the adjusted, unadjusted and intercept models. A two-sided p value of<0.05 was considered significant.

### Process evaluation

During the planned regular meetings with members of the local ASB teams to report results and feedback, we will evaluate the introduced interventions. By exchanging thoughts and experiences on the implemented interventions, we will be able to determine what intervention(s) will work best in daily practice. For the process evaluation, we will evaluate the deimplementation strategy adherence and compliance. Therefore, all elements of the deimplementation strategy will be explored in all participating hospitals. We will award the hospitals with feedback points for all elements of the deimplementation strategy.[27] For

example, points will be given for the number of persons present at the planned meetings and for every strategy that is implemented.

### Economic evaluation

We will primarily perform a cost-effectiveness analysis, in which we evaluate the cost effectiveness ratio (CER) (CER=cost of deimplementation/healthcare cost reduction). Our goal was to achieve a reduction in urine culture, urinalysis and antimicrobial prescriptions, and thereby a reduction in costs. The costs of our deimplementation strategy will be divided in non-recurrent study-related costs and recurrent de-implementation-related costs. Non-recurrent study-related costs include material costs, developmental costs of the deimplementation strategy and costs of evaluation of the deimplementation. Recurrent costs include the costs to introduce the deimplementation strategy, such as time of the extra meetings of the ASB team and time invested by the local champion. For feasibility reasons, we will use the number of all urine cultures, urinalyses and antimicrobial prescriptions, representing the main cost benefits due to our strategy. The costs for urine cultures, urinalyses and antimicrobials used in the analyses will be based on Dutch guideline on healthcare economic evaluations.[28] The result of the cost-effectiveness analysis will be used in a budget impact analysis to assess the financial impact on the hospital's budget to implement and sustain the deimplementation strategy.

## ETHICS AND DISSEMINATION

Ethical approval was obtained on 13 December 2019 from the medical ethics research committee of the Academic Medical Centre (Amsterdam, the Netherlands), with a waiver for informed consent. Local feasibility was obtained by the local institutional review boards of all participating hospitals. This trial is registered at Netherlands Trial Register, trialregister.nl/trial/8242. All relevant results of our study will be disseminated through publications in peer-reviewed journals and presentations at relevant (scientific) conferences. No identifiable patient data will be disseminated.

### Patient and public involvement

This research will be performed without patient involvement. Patients will not be invited to comment on the study design and will not be consulted to develop patient-relevant outcomes or interpret the results. Patients will not be invited to contribute to the writing or editing of this document for readability or accuracy.

### Data availability

Data collected from this study, including deidentified individual participant data, will be made available on publication to investigators whose proposed use of the data has been approved by an independent review committee identified for this purpose. Proposals should be directed to the chief investigator SEG (SE.geerlings@

amsterdamumc.nl); to gain access, data requestors will need to sign a data access agreement.

## DISCUSSION

This study protocol describes the design, deimplementation strategy and process evaluation to reduce overtreatment of ASB in the daily practice of emergency medicine. Our deimplementation strategy could prevent the use of unnecessary urine diagnostics and subsequent inappropriate treatment of ASB, improve healthcare quality, lower the increase in antimicrobial resistance and save costs. If proven to be (cost) effective, this study could provide a nationwide strategy to reduce overtreatment of ASB.

In our study, we will not evaluate the clinical outcome of our strategy, including potential clinical benefits, such as reduced side effects of antibiotics, since previous studies have already shown that ASB in non-risk patients does not lead to an increased risk of symptomatic UTI and that it is safe to not treat ASB.[29 30]

By using a stepped-wedge cluster randomised design, we will be able to compare the outcomes within a cluster between the time intervals in which a cluster received the control and the interventions. Furthermore, our intervention will be enrolled in a systematic manner and eventually will be introduced in all participating hospitals. A novel part of our study compared with previous studies that aimed to reduce overtreatment of ASB is that we will be using local champions in each hospital and adjust the intervention based on the local situation.

A limitation of stepped wedge designs is that this requires larger sample sizes, as patients' characteristics may be similar within one cluster. By randomisation of the start time of the clusters, we intend to balance these important characteristics. However, this cannot be ensured with our relatively small number of clusters.[31] Another possible limitation is that we will not prospectively assess clinical symptoms ourselves and are limited to documentation of symptomatology in patient files. This might overestimate the number of patients with asymptomatic bacteriuria in the baseline and intervention periods. Since this should be similar in the baseline and intervention periods, we do not expect that this will influence our results. Furthermore, we cannot evaluate the impact of an individual intervention. However, we expect that our deimplementation strategy is low cost and well suitable for broad-scale implementation.

## Author affiliations
[1]Internal Medicine, Infectious Diseases, Amsterdam UMC, Locatie AMC, Amsterdam, North Holland, The Netherlands
[2]Internal Medicine, Infectious Diseases, OLVG Locatie Oost, Amsterdam, Noord-Holland, The Netherlands
[3]Internal Medicine, Infectious Diseases, HagaZiekenhuis, Den Haag, Zuid-Holland, The Netherlands
[4]Emergency Medicine, Amsterdam UMC, Locatie AMC, Amsterdam, North Holland, The Netherlands
[5]Medical Microbiology, Amsterdam UMC, Locatie AMC, Amsterdam, North Holland, The Netherlands

**Acknowledgements** We thank Vanessa C Harris (physician, Infectious Diseases, Internal Medicine, Amsterdam UMC, University of Amsterdam, Netherlands), for proofreading and spell checking the manuscript.

**Contributors** BJL and SEG were involved in the conception of the study and coordinated the funding application. TMZXKvH, SEG and BJL drafted the manuscript. BJL outlined the statistical analysis with support of the Clinical Research Unit of the Academic Medical Centre (Amsterdam, the Netherlands). JES, CvN, JBS, and CS revised the manuscript critically for important intellectual content. All authors read and approved the final version of the manuscript.

**Funding** This study is supported by the Netherlands Organisation for Health Research and Development (ZonMw) grant number 839 205 002.

**Competing interests** None declared.

**Patient and public involvement** Patients and/or the public were not involved in the design, conduct, reporting or dissemination plans of this research.

**Patient consent for publication** Not required.

**Provenance and peer review** Not commissioned; externally peer reviewed.

**ORCID iDs**
Tessa MZXK van Horrik http://orcid.org/0000-0002-3496-2203
Bart J Laan http://orcid.org/0000-0002-2357-8871

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
