## [Reviewer comments · BMJ Open]

ARTICLE DETAILS

TITLE (PROVISIONAL)	De-implementation strategy to reduce overtreatment of asymptomatic bacteriuria: a study protocol for a stepped-wedge cluster randomized trial
AUTHORS	van Horrik, Tessa; Geerlings, Suzanne; Stalenhoef, Janneke; van Nieuwkoop, Cees; Saanen, Joppe; Schneeberger, Caroline; Laan, Bart

VERSION 1 – REVIEW

REVIEWER	Lindsay A. Petty University of Michigan, United States of America
REVIEW RETURNED	11-Jun-2020

GENERAL COMMENTS	This is a protocol of a step-wedge cluster randomized trial for de-implementation of testing and treatment of ASB in the ED. Overall, this is an important topic, given that ASB is very common, and inappropriate treatment of ASB is common, and few studies thus far have demonstrated a decrease in antibiotic use with interventions aimed at decreasing culturing. Targeting the ED, as opposed to the general wards, is also a strength of this study. However, in order for the study to be repeated, I think it is important to share (in appendices or supplements) the educational material you plan to use at the initiation of the intervention at each site. In addition, the other novel part of this study is the use of the “ikea effect”, as they will be using local hospital champions (and a multidisciplinary group including RN), to get feedback and adjust the intervention locally. This is the really novel part of the ASB intervention, and needs to be highlighted more. Major concerns: • Depending on the volume in the EDs, I would anticipate more than 14 ASB patients during the study period at each site. Are you going to stop collecting after you reach that number, or you are anticipating that is the minimum you need, but you will keep collecting throughout the study period? It may be helpful to have a statistician review the power calculations to ensure that this will really be enough patients to power the study appropriately, in particular since each site will have variations given the local adaptations planned.• Have you considered how to approach patients with altered mental status and possible systemic signs of infection? In the most updated IDSA guidelines 2019, it was stated that antibiotics may be reasonable in this case. You may want to consider a subgroup analysis of those with AMS and leukocytosis, or 2 or more SIRS criteria, or hypotension. Those are patients that the clinical decision to withhold antibiotics, in the setting of them not being able to describe symptoms, may be more reasonable.
--

	Minor concerns:  • Line 11 on page 4: informal wording, would re-phrase • Line 38-42 on page 4: sentence doesn't read correctly for the conclusion of the study • Multiple (minor) grammatical errors, would benefit from further editing. • Table 1: fever should also be "with no other recognized cause" • Why are you planning to adjust for positive urinalysis if you are excluding all negative urinalyses? And for other associated risk factors, what about dementia, incontinence, and peripheral leukocytosis? • How do you plan to calculate the "costs to introduce the de-implementation strategy"? Are you taking into account the time of the hospital champions? • The lack of assessment of clinical outcomes is a weakness, although I recognize that you really won't be able to do this as you will not have a way to recognize patients who did not have the test sent. • Figure 2: would label those patients who were not given antibiotics for ASB (currently labeled "no treatment"), instead as "No antimicrobials given" • Another limitation to highlight is that you will be potentially overestimating patients with ASB, as you will be limited by documentation • I would also recommend including "reason for urine testing documented in the medical record" as something to abstract from each chart
--	---

REVIEWER	Sergio Alejandro Gómez Ochoa Universidad Industrial de Santander, Colombia
REVIEW RETURNED	12-Jun-2020

GENERAL COMMENTS	De-implementation strategy to reduce overtreatment of asymptomatic bacteriuria: a study protocol for a stepped wedge cluster randomized trial Peer review comments Van Horrik and colleagues designed a stepped wedge cluster randomized trial to evaluate the efficacy of a de-implementation strategy to reduce overtreatment of asymptomatic bacteriuria in the emergency departments of five hospitals in the Netherlands. Their objective is to reduce the inappropriate treatment of ASB with antimicrobials by 50%. They will also evaluate the impact of the intervention in the duration of antibiotic therapy for ASB, the number of urine cultures (UC) and urine analyses (UA) performed per 1000 patients, and the trend of the treatment of positive UA in asymptomatic patients. The strategy has a strong basis of evidence to support it, and the design of the study is, in general terms, adequate. I have the following minor concerns:  1. Please adapt the definition of ASB in Table 1 to the complete IDSA one, clarifying the definition by sex. It needs to be mentioned as well the definition according to each method of sample collection (indwelling catheter vs. voided urine specimen). 2. Will you exclude participants in which UTI symptoms cannot be ruled out? For example, individuals with neurologic or neurodegenerative diseases or patients with stroke sequelae who cannot clearly define ASB symptoms.
---

	3. Did you consider a transition period before the inclusion of every new cluster? It would be relevant as these types of strategies are a process. Physicians and, in general, all health care workers struggle to leave old manners and replace them with new evidence-based conduct. This period of transition would allow the delivery of the strategy to the target population while not “contaminating” the results of the study because of the initial problems with adherence to the strategy. The adjustment by temporal trend may mitigate this impact, but it is not optimal. 4. The strategy section would undoubtedly benefit from a more detailed description of the way it is going to be delivered. Who is going to lead the initial educational meetings? Which specific themes are going to be discussed in each one? Each local champion will decide the way he/she is going to carry out the strategy? 5. For the analysis, it would be interesting to also evaluate the pathogen species as a factor influencing the decision of treating. Would the intervention work for all microbes? (In the study of Flokas et al., the isolation of gram-negative bacteria was identified as a risk factor for inappropriate ASB treatment). This data is easy to collect. 6. As you know, reporting guidelines specific to stepped wedge cluster randomized trials do not exist. However, the study of Hemming et al. (I will leave the reference below) recommended a series of modifications to the Consort 2010 cluster extension for reporting of stepped wedge cluster randomized trials. It may be the most accurate guideline available. Hemming K, Haines TP, Chilton PJ, Girling AJ, Lilford RJ. The Stepped Wedge Cluster Randomised Trial: Rationale, Design, Analysis, and Reporting. BMJ. 2015 6;350:h391.
--	---

REVIEWER	Emily Rousham Loughborough University, UK
REVIEW RETURNED	17-Jun-2020

GENERAL COMMENTS	The study protocol describes an important intervention to reduce unnecessary prescription of antibiotics for asymptomatic bacteriuria. The study design is appropriate and ethical considerations have been addressed. Major comment I am only familiar with the UK guidance of UTI, and these probably differ to those in the Netherlands. However, clinical guidelines for diagnosis are very different for adults over 65 years, and urinalysis (dipsticks) should not be used for any older adults. https://assets.publishing.service.gov.uk/government/uploads/system/uploads/attachment_data/file/755889/PHE_UTI_flowchart_-_over_65.pdf I wonder if the team might consider a separate flow chart/algorithm for intervention materials that is specifically for adults over 65 years? If this was already considered, but not adopted in the proposed intervention, the justification for this could be provided in the manuscript. There is increasing evidence in the literature against the use of dipsticks for any patient over the age of 65 years. Other minor suggestions 1. Could the authors briefly explain the context of the listed campaigns ‘Choosing Wisely’ and ‘Do not Do’, are these for UTI or other health treatments? Similarly, briefly expand on ‘To Do or Not to Do’? What healthcare treatment/issues does this refer to?
--

	2. The introduction underplays the age-related prevalence of ASB, this is a much greater risk in adults over 65 years – some relevant literature could be included. 3. Some brief country context for the intervention studies reviewed in the introduction section could be included, some are in the Netherlands, some are elsewhere. 4. Page 10, Process Evaluation – was a qualitative component to the process evaluation considered? Qualitative factors rather than quantitative measures may play an important role in the trial success, for example qualitative indicators might provide insights into the role of a local champion and usefulness of educational materials etc.
--	---

REVIEWER	
REVIEW RETURNED	

GENERAL COMMENTS	
--

REVIEWER	
REVIEW RETURNED	

GENERAL COMMENTS	
--

VERSION 1 – AUTHOR RESPONSE

	Reviewer #1		
6	This is a protocol of a step-wedge cluster randomized trial for de-implementation of testing and treatment of ASB in the ED. Overall, this is an important topic, given that ASB is very common, and inappropriate treatment of ASB is common, and few studies thus far have demonstrated a decrease in antibiotic use with interventions aimed at decreasing culturing. Targeting the ED, as opposed to the general wards, is also a strength of this study. However, in order for the study to be repeated, I think it is important to share (in appendices or supplements) the educational material you plan to use at the initiation of the intervention at each site. In addition, the other novel part of this study is the use of the “ikea effect”, as they will be using local hospital champions (and a multidisciplinary group including RN), to get feedback and adjust the intervention locally. This is the really novel part of the ASB intervention, and needs to be highlighted more.	We thank the reviewer for her comment and have, as suggested, added the remark of the novel part of the ASB intervention to the manuscript in the Discussion Section on page 11-12 “A novel part of our study compared to previous studies that aimed to	No changes made

		reduce overtreatment of ASB is that we will be using local champions in each hospital and adjust The educational material has to be developed yet and will be published with the main study results the intervention based on the local situation.”	
7	Depending on the volume in the EDs, I would anticipate more than 14 ASB patients during the study period at each site. Are you going to stop collecting after you reach that number, or you are anticipating that is the minimum you need, but you will keep collecting throughout the study period? It may be helpful to have a statistician review the power calculations to ensure that this will really be enough patients to power the study appropriately, in particular since each site will have variations given the local adaptations planned.	We agree and understand your concern. We initially calculated our sample size together with a statistician and we anticipated that the number would be the minimum we need and we were planning to continue collecting	Added clarification to sample size section, page 10: “...a total of at least 420 patients ...”

		after we reach the number of a study period. In addition, the prevalence of ASB in the Netherlands is not known. Therefore, to calculate our sample size, we used a pooled prevalence of inappropriate antimicrobial treatment for ASB of 45%, which was the result of a meta-analysis of Flokas et al. in 2017. Since the prevalence of inappropriate treatment of ASB is unknown, the calculated sample size is a minimum and we will keep collecting data during the study period after reaching	
--	--	--	--

		14 patients at each site.	
8	Have you considered how to approach patients with altered mental status and possible systemic signs of infection? In the most updated IDSA guidelines 2019, it was stated that antibiotics may be reasonable in this case. You may want to consider a subgroup analysis of those with AMS and leukocytosis, or 2 or more SIRS criteria, or hypotension. Those are patients that the clinical decision to withhold antibiotics, in the setting of them not being able to describe symptoms, may be more reasonable.	We will not define this group as asymptomatic, but as patients with systemic symptoms of an infection and consider treatment with antimicrobials appropriate in this case. Since this group is considered as patients with symptomatic symptoms, we will exclude them.	No changes made.
9	Line 11 on page 4: informal wording, would re-phrase	We assume this comment relates to the last sentence of the introduction of the abstract; "The main objective of our study is to reduce overtreatment of ASB by 50% by using a de-	We rephrased the sentence of the abstract on page 2: "We have designed a study protocol with the main objective of reducing overtreatment of ASB by 50%

		implemen tation strategy.”	through use of a de- impleme ntation strategy.”
1 0 .	Line 38-42 on page 4: sentence doesn't read correctly for the conclusion of the study	We assume this comment relates to the “ethics and dissemina tion” part of the abstract and agree with this comment and changed the sentence. “Our study aims to reduce inappropri ate screening (...) institution al review boards of all participati ng hospitals.”	We rephras ed the “ethics and dissemi nation” part of the abstract on page 2: “Ethical approval was obtained from Medical Ethics Researc h Committ ee of the Academi c Medical Centre, Amsterd am the Netherla nds with a waiver for informed consent. Local feasibilit y will be obtained by the local institutio nal review boards of all participa ting hospital s. Our study aims to reduce

			inappropriate screening and treatment of ASB in emergency departments, improve healthcare quality, lower the increase in antimicrobial resistance, and save costs. If proven (cost)-effective, this study provides a well suited strategy for a nationwide approach to reduce overtreatment of ASB.”
1 1 .	Multiple (minor) grammatical errors, would benefit from further editing.	As suggested the whole manuscript was checked and changed by a native English speaker.	We corrected multiple grammatical errors of the manuscript. All changes are marked.
1 2 .	Table 1: fever should also be “with no other recognized cause”	We used the definition	We added “with no

		of a symptomatic UTI according to the CDC, which mentions "Fever is a non-specific symptom of infection and cannot be excluded from UTI determination because it is clinically deemed due to another recognized cause."	other recognized cause" to fever in Table 1 on page 6-7.
1 3 .	Why are you planning to adjust for positive urinalysis if you are excluding all negative urinalyses? And for other associated risk factors, what about dementia, incontinence, and peripheral leukocytosis?	We do not exclude all negative urinalyses or not performed urinalyses if there is a positive urine culture. We are planning to adjust for positive urinalyses , since we expect that these may influence the decision about starting antibiotics . We expect that	No changes made.

		physicians are more likely to prescribe antibiotics for a positive result, without or before performing a urine culture. We will adjust for the most important risk factors. This was discussed with our statistician, as stated on page 10. However, we will also collect data on dementia and peripheral leucocytosis.	
14.	How do you plan to calculate the “costs to introduce the de-implementation strategy”? Are you taking into account the time of the hospital champions?	We agree that this is not completely clear in the protocol. We did not plan to account for the time of the hospital champions, because the champions would raise	Revised economic evaluation on page 11: “Recurrent costs..., such as time of the extra meetings of the ASB-teams and time invested by local champion”.

		awareness during regular meetings. We will take into account the time of the extra meetings of the ASB-team (including time of the local champion), since this takes place next to their regular work. Next to this, we will include the time of educational meetings and costs for pocket cards and posters.	
15	The lack of assessment of clinical outcomes is a weakness, although I recognize that you really won't be able to do this as you will not have a way to recognize patients who did not have the test sent.	We agree that this is a weakness. However, we decided not to assess clinical outcomes, since the results of previous studies already showed that ASB in non-risk patients does not lead to	No changes made.

		increased risk of symptomatic UTI and that it is safe to not treat ASB.	
16	Figure 2: would label those patients who were not given antibiotics for ASB (currently labeled “no treatment”), instead as “No antimicrobials given”	We agree and changed the label.	We changed the label in figure 2 from “No treatment” to “No antimicrobials given”.
17	Another limitation to highlight is that you will be potentially overestimating patients with ASB, as you will be limited by documentation.	We do not know whether this is a limitation, since this will be similar in the baseline and intervention period. Therefore, we do not expect that this will influence the results.	We added this possible limitation to the ‘strengths and limitations’ part on page 3 and to the ‘discussion’ part on page 12.
18	I would also recommend including “reason for urine testing documented in the medical record” as something to abstract from each chart	This item was already included in our data collection plan, however, we did not mention it specifically in the study protocol.	We added a sentence to the paragraph “Data collection” on page 7 “We will also collect the reasons for urine testing documented in

			the medical record.”
	Reviewer #2		
	Van Horrik and colleagues designed a stepped wedge cluster randomized trial to evaluate the efficacy of a de-implementation strategy to reduce overtreatment of asymptomatic bacteriuria in the emergency departments of five hospitals in the Netherlands. Their objective is to reduce the inappropriate treatment of ASB with antimicrobials by 50%. They will also evaluate the impact of the intervention in the duration of antibiotic therapy for ASB, the number of urine cultures (UC) and urine analyses (UA) performed per 1000 patients, and the trend of the treatment of positive UA in asymptomatic patients. The strategy has a strong basis of evidence to support it, and the design of the study is, in general terms, adequate. I have the following minor concerns:		
19.	Please adapt the definition of ASB in Table 1 to the complete IDSA one, clarifying the definition by sex. It needs to be mentioned as well the definition according to each method of sample collection (indwelling catheter vs. voided urine specimen).	We thank the reviewer for his comment. However, the most recent IDSA guideline of 2019 stated that one urine culture will suffice for diagnosing ASB. Therefore, we did not change our definition of ASB in table 1.	No changes made.
20.	Will you exclude participants in which UTI symptoms cannot be ruled out? For example, individuals with neurologic or neurodegenerative diseases or patients with stroke sequelae who cannot clearly define ASB symptoms.	We thank the reviewer for his comment. We will not exclude all patients with neurologic disease, because they can also have ASB or systemic symptoms and decided	No changes made.

		only to exclude patients with systemic symptoms of a UTI following the clinical decision of the attending physician.	
2 1 .	Did you consider a transition period before the inclusion of every new cluster? It would be relevant as these types of strategies are a process. Physicians and, in general, all health care workers struggle to leave old manners and replace them with new evidence-based conduct. This period of transition would allow the delivery of the strategy to the target population while not “contaminating” the results of the study because of the initial problems with adherence to the strategy. The adjustment by temporal trend may mitigate this impact, but it is not optimal.	We understand the reviewer’s comment and did consider a transition period. However, we also consider the strategy as a process. The start of the intervention is the kick-off meeting and all activities thereafter are part of this process. Therefore, we did not include a transition period before the inclusion of every new cluster.	No changes made.
2 2 .	The strategy section would undoubtedly benefit from a more detailed description of the way it is going to be delivered. Who is going to lead the initial educational meetings? Which specific themes are going to be discussed in each one? Each local champion will decide the way he/she is going to carry out the strategy?	We agree that a more detailed description of the	We added a more detailed description to the

		strategy could be delivered. However, the strategies to de-implementation of ASB depend on the results in the baseline period and will be focused on the local situation in the participating hospital. We expect that these results are different in the participating hospital, which makes it difficult to provide a more detailed description in this study protocol. We have planned to provide a detailed description in the final article.	strategy section on page 8: "During these educational meetings, we will inform all healthcare workers about the current ASB guidelines and discuss the local barriers and facilitators to reduce its over-treatment." and "The kick-off meeting will be led by the project coordinator and the local champion".
23.	For the analysis, it would be interesting to also evaluate the pathogen species as a factor influencing the decision of treating. Would the intervention work for all microbes? (In the study of Flokas et al., the isolation of gram-negative bacteria was identified as a risk factor for	We agree that it would be interesting	We added this item to the

	inappropriate ASB treatment). This data is easy to collect.	g to also evaluate the pathogen species as a factor influencing the decision of treating and have planned to collect these data.	data collection part on page 7 “For patients with ASB, we will collect data on antimicrobial treatment, including antimicrobial prescription and duration of antimicrobial therapy, and the specified bacterial count and pathogen according to the standard operating procedure of the microbiology laboratory.”
2 4 .	As you know, reporting guidelines specific to stepped wedge cluster randomized trials do not exist. However, the study of Hemming et al. (I will leave the reference below) recommended a series of modifications to the Consort 2010 cluster extension for reporting of stepped wedge cluster randomized trials. It may be the most accurate guideline available. Hemming K, Haines TP, Chilton PJ, Girling AJ, Lilford RJ. The Stepped Wedge Cluster Randomised Trial: Rationale, Design, Analysis, and Reporting. BMJ. 2015 6;350:h391.	We have read the suggested article about reporting guidelines for stepped wedge cluster randomized trials and have followed	We added a sentence to the “Methods and analysis – study design and setting” on page 5. “For the develop

		their recommendations (Table 2) during the development of our study protocol.	ment of our study protocol, we used the reporting guidelines of stepped wedge cluster randomised trials. ¹⁷ ”
	Reviewer #3		
	The study protocol describes an important intervention to reduce unnecessary prescription of antibiotics for asymptomatic bacteriuria. The study design is appropriate and ethical considerations have been addressed.		
2 5 .	I am only familiar with the UK guidance of UTI, and these probably differ to those in the Netherlands. However, clinical guidelines for diagnosis are very different for adults over 65 years, and urinalysis (dipsticks) should not be used for any older adults. https://assets.publishing.service.gov.uk/government/uploads/system/uploads/attachment_data/file/755889/PHE_UTI_flowchart_-_over_65.pdf I wonder if the team might consider a separate flow chart/algorithm for intervention materials that is specifically for adults over 65 years? If this was already considered, but not adopted in the proposed intervention, the justification for this could be provided in the manuscript. There is increasing evidence in the literature against the use of dipsticks for any patient over the age of 65 years.	We agree that a dipstick is frequently false positive in adults over 65 years. However, a negative dipstick could still rule out a UTI. Our flowchart is based on both international and Dutch guidelines for ASB and urinary tract infections. One of the Dutch guidelines (Verenso) is conducted specifically for adults over 65 years.	No changes made.

		However, in the Dutch guidelines of UTI, it is not stated that we should not use a dipstick for patients over the age of 65 years. Therefore, we did not create a separate flowchart specifically for adults over 65 years.	
2 6 .	Could the authors briefly explain the context of the listed campaigns 'Choosing Wisely' and 'Do not Do', are these for UTI or other health treatments? Similarly, briefly expand on 'To Do or Not to Do'? What healthcare treatment/issues does this refer to?	As stated in the introduction, these campaigns have been launched to reduce unnecessary care. These international campaigns address all kinds of health treatments and diagnostics.	We added this information briefly in the introduction on page 4: "This is exemplified by international campaigns, such as Choosing Wisely and Do not Do prompts, that were launched to reduce unnecessary care in several medical

			services , resulting in less risks for the patients, saving costs and increasi ng value for care.”
2 7 .	The introduction underplays the age-related prevalence of ASB, this is a much greater risk in adults over 65 years – some relevant literature could be included.	We agree that the age-related prevalence of ASB was underplayed and added a relevant reference to the manuscript (Nicolle LE. Asymptomatic bacteriuria: when to screen and when to treat. Infectious disease clinics of North America 2003;17(2):367-94. [published Online First: 2003/07/10]).	We added a sentence in the introduction to emphasize the age-related prevalence of ASB, page 4: “The prevalence of ASB in adults over 65 years is high, since the prevalence of ASB increases with age.”
2 8 .	Some brief country context for the intervention studies reviewed in the introduction section could be included, some are in the Netherlands, some are elsewhere.	We agree that some brief country context for the intervention studies that are mentioned in the	We added brief country context for the intervention studies named in the

		introduction is applicable	introduction, page 4-5.
29	Page 10, Process Evaluation – was a qualitative component to the process evaluation considered? Qualitative factors rather than quantitative measures may play an important role in the trial success, for example qualitative indicators might provide insights into the role of a local champion and usefulness of educational materials etc.	We agree that qualitative factors could add important information to the process evaluation. Unfortunately, we have limited resources and therefore we are unable to perform a qualitative analysis for the process evaluation, such as in depth interviews to get insight in the role of the local champion. Therefore, we will perform a practical process evaluation to investigate the correlation between the adherence to the de-implementation strategy and reduction	No changes made.

		of overtreatment of ASB. (See our earlier study: Laan BJ, Maaskant JM, Spijkerman IJB, et al. De-implementation strategy to reduce inappropriate use of intravenous and urinary catheters (RICAT): a multicentre, prospective, interrupted time-series and before and after study. The Lancet Infectious diseases 2020 doi: 10.1016/S1473-3099(19)30709-1 [published Online First: 2020/03/11]).	
--	--	---	--

VERSION 2 – REVIEW

REVIEWER	Lindsay A. Petty University of Michigan, United States of America
REVIEW RETURNED	25-Aug-2020
GENERAL COMMENTS	• Overall, I think the edits made in response to all 3

	reviewers were great, and I have only a few minor comments/clarifications at this point.  • The Flokas study citing inappropriate abx treatment of 45% is one study. However, other published studies have demonstrated a higher rate of inappropriate treatment of ASB (of which you cite some later in the manuscript), so you may want to give a range of inappropriate ASB treatment in the abstract. Some references below.  o Spivak ES, Burk M, Zhang R, et al; Management of Urinary Tract Infections Medication Use Evaluation Group. Management of bacteriuria in Veterans Affairs hospitals. Clin Infect Dis. 2017;65(6):910-917. o Hartley S, Valley S, Kuhn L, et al. Overtreatment of asymptomatic bacteriuria: identifying targets for improvement. Infect Control Hosp Epidemiol. 2015;36(4):470-473. o D'Agata E, Loeb MB, Mitchell SL. Challenges in assessing nursing home residents with advanced dementia for suspected urinary tract infections. J Am Geriatr Soc. 2013;61(1):62-66. o Petty LA, Vaughn VM, Flanders SA, et al. Risk Factors and Outcomes Associated With Treatment of Asymptomatic Bacteriuria in Hospitalized Patients. JAMA Intern Med. 2019;179(11):1519–1527. • For my prior comment, “Line 11 on page 4: informal wording, would re-phrase”, I was referring to the line 11 on page 4 in the PDF, which was: “In the past years, international campaigns, such as Choosing Wisely and Do not Do prompts, have been launched to reduce unnecessary care, resulting in less risks for the patients, saving costs and giving room for valuable care.” You made changes based on another reviewers comments, and this now reads well. Thank you. • I asked why you were planning to exclude all negative urinalyses. You responded: “We do not exclude all negative urinalyses or not performed urinalyses if there is a positive urine culture. We are planning to adjust for positive urinalyses, since we expect that these may influence the decision about starting antibiotics. We expect that physicians are more likely to prescribe antibiotics for a positive result, without or before performing a urine culture.” I’m sorry, but I’m still confused. In “patient selection”, you write, “We will screen all patients (≥ 18 years old) who have urine tests (culture and/or urinalysis) that were obtained during an ED presentation (Figure 2). First, we will exclude all negative urinalyses (defined as absence of leukocyte esterase and nitrite in dipsticks, and absence of leukocytes and bacteria in microscopic analysis) and negative urine cultures (Table 1).” To me, this means you will exclude negative urinalyses. I would agree with your plan to include negative urinalyses, and adjust for positive urinalyses, as you responded, but the manuscript does not read this way to me yet. • In response to my question regarding patients with AMS, you wrote, “We will not define this group as asymptomatic, but as patients with systemic symptoms of an infection and consider treatment with antimicrobials appropriate in this case. Since this group is considered as patients with symptomatic symptoms, we will exclude them.” Do you mean that you will exclude all patients with AMS, or those patients with AMS and systemic signs of a possible infection? I think you mean the later (those with systemic symptoms), which I would agree with, but can you please clarify this then in the protocol?
--	--

REVIEWER	Sergio Alejandro Gómez Ochoa Fundación Cardiovascular de Colombia
-----------------	--

	Universidad Industrial de Santander
REVIEW RETURNED	18-Aug-2020

GENERAL COMMENTS	All the comments have been addressed. I have no additional recommendations.
---

REVIEWER	Emily Rousham Loughborough University, UK
REVIEW RETURNED	10-Aug-2020

GENERAL COMMENTS	The revisions address the main points raised in the review.
---

VERSION 2 – AUTHOR RESPONSE

Item	Editor's and reviewer's comments	Author's response	Changes in revised paper
	Reviewer #1 Lindsay A. Petty University of Michigan, United States of America		
	Overall, I think the edits made in response to all 3 reviewers were great, and I have only a few minor comments/clarifications at this point.		
1.	The Flokas study citing inappropriate abx treatment of 45% is one study. However, other published studies have demonstrated a higher rate of inappropriate treatment of ASB (of which you cite some later in the manuscript), so you may want to give a range of inappropriate ASB treatment in the abstract. Some references below.  o Spivak ES, Burk M, Zhang R, et al; Management of Urinary Tract Infections Medication Use Evaluation Group. Management of bacteriuria in Veterans Affairs hospitals. Clin Infect Dis. 2017;65(6):910-917. o Hartley S, Valley S, Kuhn L, et al. Overtreatment of asymptomatic bacteriuria: identifying targets for improvement. Infect Control Hosp Epidemiol. 2015;36(4):470-473. o D'Agata E, Loeb MB, Mitchell SL. Challenges in assessing nursing home residents with advanced dementia for suspected urinary tract infections. J Am Geriatr Soc. 2013;61(1):62-66. o Petty LA, Vaughn VM, Flanders SA, et al. Risk Factors and Outcomes Associated With Treatment of Asymptomatic Bacteriuria in Hospitalized Patients. JAMA Intern Med. 2019;179(11):1519–1527. 	We have read the suggested references and agree that a range of inappropriate ASB treatment in the abstract would be more complete. Although, D'Agata et al. performed their study in nursing homes and since our study setting is the emergency department in hospitals, we consider this reference as less applicable to our study. References: Flokas: pooled prevalence of inappropriate ASB treatment: 45% (range 16.24-76.67%). Spivak: 494/729 (67.8%) patients with ASB + antibiotic treatment. Setting: veterans affairs hospitals in the USA. Hartley: 153 patients with positive urine culture AND treatment with antimicrobials within 72h of urine culture collection of which 94 (61.4%) had ASB → 60 patients (63.8%) treated with antimicrobials for UTI. Setting: University hospital in Michigan, USA.	We added a range of inappropriate ASB treatment in the abstract on page 3 'introduction': "Earlier studies have shown that the prevalence of this inappropriate treatment ranges from 45 to 83%."

		D'Agata: 110/131 patients (84%) NO minimal criteria to initiate antimicrobials → 82/110 (74.5%) antimicrobials given → 56 positive urinalysis + cultures. Setting: nursing home residents, USA. Petty: 2772/7252 (38.2%) patients with ASB → 2733 complete treatment data → 2259 (82.7%) received antibiotics which was considered inappropriate. Setting: hospitals, USA.	
2.	For my prior comment, "Line 11 on page 4: informal wording, would rephrase", I was referring to the line 11 on page 4 in the PDF, which was: "In the past years, international campaigns, such as Choosing Wisely and Do not Do prompts, have been launched to reduce unnecessary care, resulting in less risks for the patients, saving costs and giving room for valuable care." You made changes based on another reviewers comments, and this now reads well. Thank you.	Thank you.	No changes made.
3.	I asked why you were planning to exclude all negative urinalyses. You responded: "We do not exclude all negative urinalyses or not performed urinalyses if there is a positive urine culture. We are planning to adjust for positive urinalyses, since we expect that these may influence the decision about starting antibiotics. We expect that physicians are more likely to prescribe antibiotics for a positive result, without or before performing a urine culture." I'm sorry, but I'm still confused. In "patient selection", you write, "We will screen all patients (≥ 18 years old) who have urine tests (culture and/or urinalysis) that were obtained during an ED presentation (Figure 2). First, we will exclude all negative urinalyses (defined as absence of leukocyte esterase and nitrite in dipsticks, and absence of leukocytes and bacteria in microscopic analysis) and negative urine cultures (Table 1)." To me, this means you will exclude negative urinalyses. I would agree with your plan to include negative	We agree that the 'patient selection' part on page 6 was not clear on this point. Therefore, we changed the sentence in the manuscript.	We rephrased the sentence in the 'patient selection' on page 6: "First, we will exclude all patients that have no positive urinalysis or urine culture, so if both urinalysis and culture are negative (Table 1)."

	urinalyses, and adjust for positive urinalyses, as you responded, but the manuscript does not read this way to me yet.		
4.	In response to my question regarding patients with AMS, you wrote, "We will not define this group as asymptomatic, but as patients with systemic symptoms of an infection and consider treatment with antimicrobials appropriate in this case. Since this group is considered as patients with symptomatic symptoms, we will exclude them." Do you mean that you will exclude all patients with AMS, or those patients with AMS and systemic signs of a possible infection? I think you mean the later (those with systemic symptoms), which I would agree with, but can you please clarify this then in the protocol?	We indeed mean the later and have clarified this exclusion criteria in the protocol. For patients with altered mental status (AMS) with systemic symptoms of an infection AND bacteriuria, treatment with antimicrobials is appropriate when alternate sources of infections are ruled out.	We added "Patients with altered mental status (AMS) with possible systemic symptoms of an infection and bacteriuria are excluded, since treatment with antimicrobials is then considered appropriate." to the exclusion criteria in the 'patient selection' part on page 6.
	Reviewer #2 Sergio Alejandro Gómez Ochoa Fundación Cardiovascular de Colombia, Universidad Industrial de Santander		
All the comments have been addressed. I have no additional recommendations.			
	Reviewer #3 Emily Rousham Loughborough University, United Kingdom		
The revisions address the main points raised in the review.